# Structure of the magnetic excitations in the spin-1/2 triangular-lattice Heisenberg antiferromagnet $Ba_3CoSb_2O_9$

Saya Ito[1], Nobuyuki Kurita[1], Hidekazu Tanaka[1], Seiko Ohira-Kawamura[2], Kenji Nakajima[2], Shinichi Itoh[3], Keitaro Kuwahara[4] & Kazuhisa Kakurai[5]

A spin-1/2 triangular-lattice Heisenberg antiferromagnet (TLHAF) is a prototypical frustrated quantum magnet, which exhibits remarkable quantum many-body effects that arise from the synergy between spin frustration and quantum fluctuation. The ground-state properties of a spin-1/2 TLHAF are theoretically well understood. However, the theoretical consensus regarding the magnetic excitations is limited. The experimental study of the magnetic excitations in spin-1/2 TLHAFs has also been limited. Here we show the structure of magnetic excitations in the spin-1/2 TLHAF $Ba_3CoSb_2O_9$ investigated by inelastic neutron scattering. Significantly different from theoretical expectations, the excitation spectrum has a three-stage energy structure. The lowest-energy first stage is composed of dispersion branches of single-magnon excitations. The second and third stages are dispersive continua accompanied by a columnar continuum extending above 10 meV, which is six times larger than the exchange interaction $J = 1.67$ meV. Our results indicate the shortcomings of the current theoretical framework.

[1] Department of Physics, Tokyo Institute of Technology, Oh-okayama, Meguro-ku, Tokyo 152-8551, Japan. [2] Materials and Life Science Division, J-PARC Center, Tokai, Ibaraki 319-1195, Japan. [3] Neutron Science Division, Institute of Materials Structure Science, High Energy Accelerator Research Organization, Tsukuba, Ibaraki 305-0801, Japan. [4] Institute of Quantum Beam Science, Ibaraki University, Mito 310-8512, Japan. [5] Comprehensive Research Organization for Science and Society (CROSS), Tokai, Ibaraki 319-1106, Japan. Correspondence and requests for materials should be addressed to H.T. (email: tanaka@lee.phys.titech.ac.jp)

Exploring quantum many-body effects has been one of the central subjects of condensed matter physics. Low-dimensional frustrated quantum magnets provide a stage to produce notable quantum many-body effects such as spin liquids[1] and quantized magnetization[2, 3]. The simplest and prototypical frustrated quantum magnet is a spin-1/2 triangular-lattice Heisenberg antiferromagnet (TLHAF) with the nearest-neighbor exchange interaction. Since a resonating-valence-bond (RVB) spin-liquid state without a long-range magnetic ordering was proposed as the ground state of the spin-1/2 TLHAFs[4, 5], great effort has been made to elucidate the nature of their ground state. The present theoretical consensus is that the ground state is an ordered state of the 120° spin structure with a significantly reduced sublattice magnetization[6–10].

Although the zero-field ground state of spin-1/2 TLHAFs is qualitatively the same as that for the classical spin, a pronounced quantum many-body effect emerges in magnetic fields. The quantum fluctuation stabilizes an up-up-down spin state in a finite magnetic field range, giving the magnetization curve a plateau at one-third of the saturation magnetization[11–16]. The magnetization curve, which is substantially different from that for the classical spin, has been calculated precisely using various approaches[13–16]. The quantum magnetization process has been quantitatively verified by high-field magnetization measurements on $Ba_3CoSb_2O_9$, which is described as a spin-1/2 TLHAF[17, 18]. The entire magnetization curve, including a high-field quantum phase transition above the 1/3-plateau[18], has been explained quantitatively by taking the weak easy-plane anisotropy and interlayer exchange interaction into account[19, 20]. Thus, the ground-state properties of a spin-1/2 TLHAF with a uniform triangular lattice and the nearest-neighbor exchange interaction are well understood both theoretically and experimentally.

In contrast to the ground-state properties, the magnetic excitations in a spin-1/2 TLHAF are less well understood. The limited theoretical consensus for single-magnon excitations is as follows: the dispersion relation of low-energy single-magnon excitations near the magnetic Bragg point ($K$ point) is described by linear spin-wave theory (LSWT). However, in a large area of the

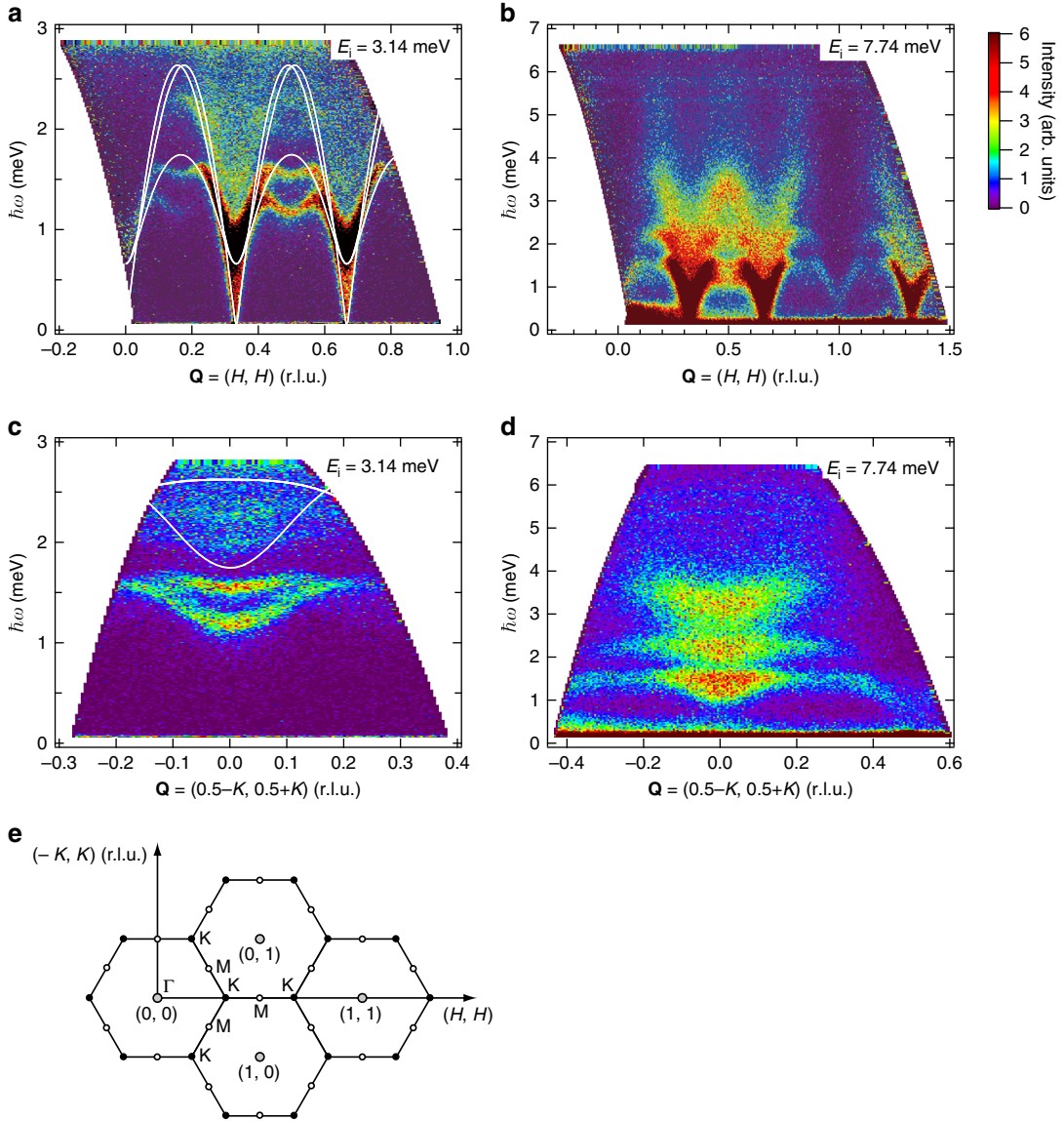

**Fig. 1** Excitation spectra of $Ba_3CoSb_2O_9$ measured at $T = 1.0$ K. **a-d** Energy-momentum maps of the scattering intensity along two high-symmetry directions $\mathbf{Q} = (H, H)$ **a**, **b** and $(-K, K)$ **c**, **d**, for varying $H$ and $K$ in reciprocal lattice units (r.l.u.), measured with incident neutron energies of $E_i = 3.14$ and $7.74$ meV. The scattering intensities were integrated over $L$ to map the scattering intensity in the 2D reciprocal lattice shown in **e**. The solid lines in **a** and **c** are dispersion curves calculated by LSWT with $J = 1.67$ meV and $\Delta = 0.046$ on the basis of the 2D model

Brillouin zone, the excitation energy is significantly renormalized downward by quantum fluctuations, causing the dispersion curve to become flat[21–26]. In addition, series expansion approach[22, 26] and fermionized-vortex theory[27] have demonstrated that the dispersion curve shows a roton-like minimum at the M point, and nonlinear spin-wave theory[23, 25] has shown that spontaneous decays of magnons occur owing to the magnon interaction, which leads to line broadening of the excitation spectrum. However, there is no theoretical consensus for the excitation continuum that reflects the characteristics of magnetic quasiparticles. The experimental study of the magnetic excitations in spin-1/2 TLHAFs has also been limited. Recently, magnetic excitations in $Ba_3CoSb_2O_9$ were investigated by inelastic neutron scattering[28, 29]. However, the energy range is limited below 3 meV and the excitation spectrum appears to be indistinct. Little is known about the excitation continuum in $Ba_3CoSb_2O_9$.

$Ba_3CoSb_2O_9$ crystallizes in a highly symmetric hexagonal structure, $P6_3/mmc$[30]. Magnetic $Co^{2+}$ ions form a uniform triangular lattice parallel to the $ab$ plane. Because the triangular layers are separated by nonmagnetic layers consisting of $Sb_2O_9$ double octahedra and $Ba^{2+}$ ions, the interlayer exchange interaction is much smaller than the intralayer exchange interaction[18]. The effective magnetic moment of $Co^{2+}$ ions with an octahedral environment can be described by the pseudospin-1/2 at low temperatures sufficiently below $|\lambda|/k_B$ 250K ($\lambda$: spin-orbit coupling constant)[17]. Because the octahedral environment of $Co^{2+}$ is close to a cubic environment in $Ba_3CoSb_2O_9$, the anisotropy of the exchange interaction is small[18]. Because of the highly symmetric crystal structure, the antisymmetric Dzyaloshinskii-Moriya interaction is absent between neighboring spins in the triangular lattice. $Ba_3CoSb_2O_9$ undergoes a magnetic phase transition at $T_N = 3.8$ K owing to the weak interlayer interaction[30]. In the ordered phase, spins lie in the $ab$ plane and form a 120° structure[18,19,29].

The effective exchange interaction between pseudospins $\mathbf{S}_i$ is described by the spin-1/2 $XXZ$ model with small easy-plane anisotropy as

$$\mathcal{H}_{ex} = \sum_{\langle i,j \rangle}^{layer} J \left( \mathbf{S}_i \cdot \mathbf{S}_j - \Delta S_i^z S_j^z \right) + \sum_{\langle l,m \rangle}^{interlayer} J' \, \mathbf{S}_l \cdot \mathbf{S}_m, \qquad (1)$$

with $0 < \Delta \ll 1$. Here, the first and second terms are the exchange interactions in the triangular layer and between layers, respectively. From the analyses of the saturation field and the collective modes observed by elecron spin resonance (ESR) measurements, the exchange parameters were evaluated to be $J = 1.67$ meV, $\Delta = 0.046$ and $J' \approx 0.12$ meV[18]. Because of the small value of $\Delta$, the exchange interaction can approximate the Heisenberg model. For simplification, the small anisotropy in the interlayer exchange interaction is neglected.

Here, we present the results of inelastic neutron scattering experiments on $Ba_3CoSb_2O_9$, which provide the whole picture of magnetic excitations in a spin-1/2 TLHAF. It is revealed that the excitation spectrum has a three-stage structure composed of single-magnon branches and two strong dispersive continua, and that the excitation continuum extends to over 10 meV that is six times larger than the exchange constant $J$.

## Results

**Two-dimensional excitation spectrum.** Figures 1a–d show energy-momentum maps of the scattering intensity along two high-symmetry directions parallel to $\mathbf{Q} = (H, H)$ and $(-K, K)$ in the two-dimensional (2D) reciprocal lattice. The scattering data were collected at 1.0 K, well below $T_N = 3.8$ K, with incident neutron energies of $E_i = 3.14$ and 7.74 meV. These two

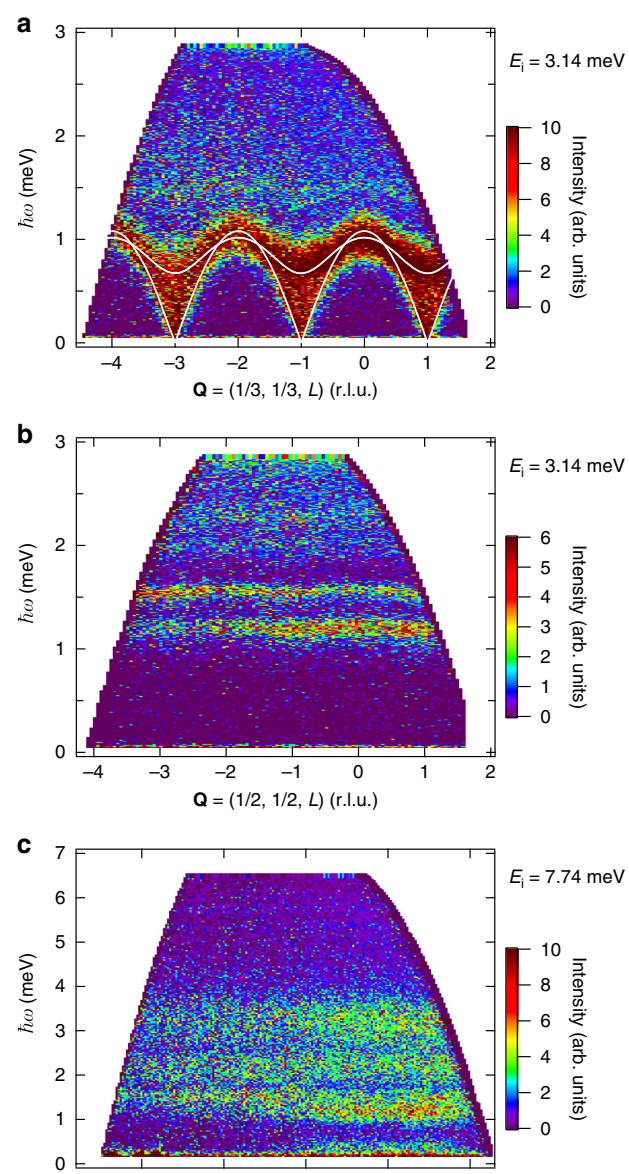

**Fig. 2** Excitation spectra of $Ba_3CoSb_2O_9$ along the $c^*$ direction. **a** Energy-momentum map of the scattering intensity along $\mathbf{Q} = (1/3, 1/3, L)$, for varying $L$, measured with $E_i = 3.14$ meV. **b**, **c** are those along $(1/2, 1/2, L)$ measured with $E_i = 3.14$ and 7.74 meV, respectively. The *solid lines* in **a** are fits calculated by LSWT with $J = 1.67$ meV, $\Delta = 0.046$ and $J' = 0.080$ meV

high-symmetry directions in the 2D reciprocal lattice are illustrated in Fig. 1e. The scattering intensities were integrated over $L$ (the wave vector along the $c^*$ direction) to map the scattering intensity in the 2D reciprocal lattice, assuming good two-dimensionality, as shown below. Two weak $\mathbf{Q}$-independent spectra between 5 and 6 meV in Fig. 1b, d are extrinsic spectra, which stem from γ-rays emitted by the collision of neutrons with $E_i = 4.68$ meV to objects made of cadmium or boron in the beam line.

Figures 2a–c show energy-momentum maps of the scattering intensity along $\mathbf{Q} = (1/3, 1/3, L)$ and $(1/2, 1/2, L)$ measured with $E_i = 3.14$ and 7.74 meV. The low-energy excitations for $\mathbf{Q} = (1/3, 1/3, L)$ are dispersive, while all the excitations for $\mathbf{Q} = (1/2, 1/2, L)$ are almost independent of $L$. This indicates that the interlayer exchange interaction is small and does not affect the

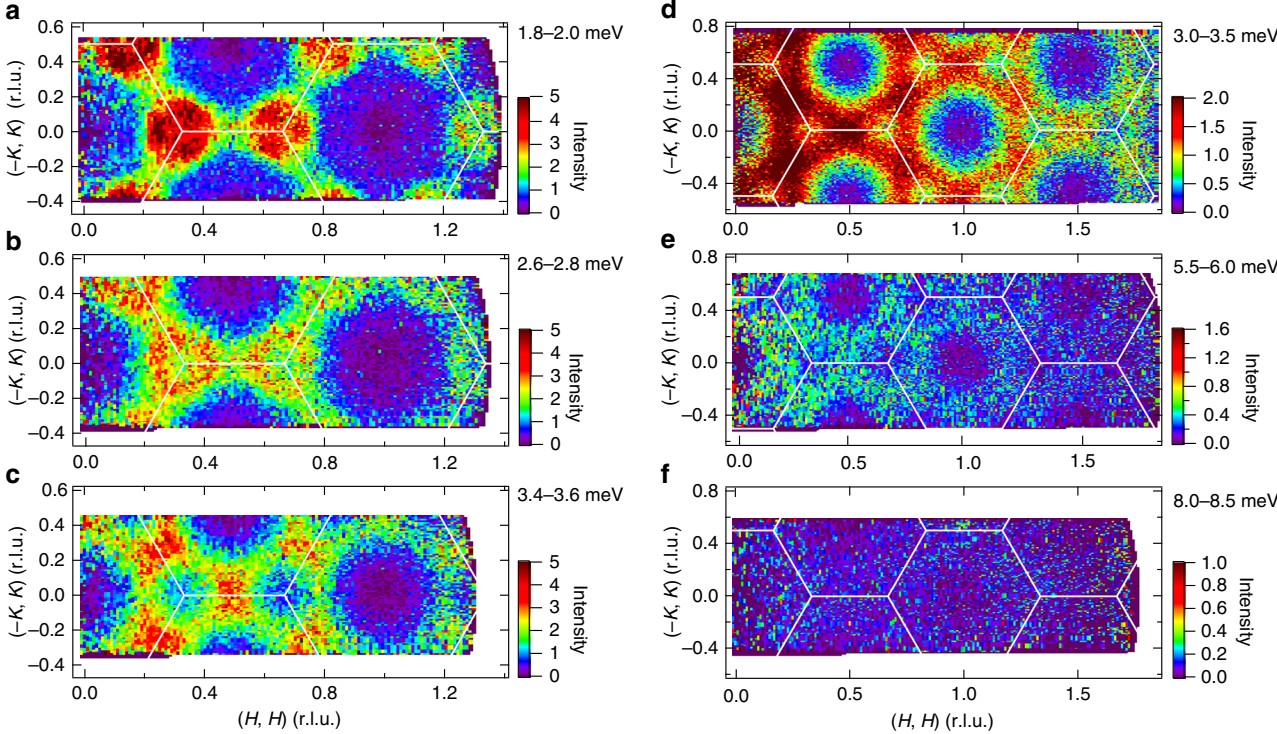

**Fig. 3** Constant-energy slices of scattering intensity. **a-c** and **d-f** are the results obtained using the neutron scattering data measured with $E_i = 7.74$ and 15.16 meV, respectively. The x and y axes are the components of **Q** parallel to (H, H) and (−K, K), respectively. Integrated energy ranges are shown in the figures. *Solid white lines* are Brillouin zone boundaries

excitations above 1 meV. Because the low-energy excitations in the vicinity of the K point can be described by LSWT, as shown below, we evaluate the interlayer exchange interaction $J'$ by applying LSWT to the dispersion curves of the single-magnon excitations for **Q** = (1/3, 1/3, L). The *solid lines* in Fig. 2a are fits with $J' = 0.080$ meV, with J and $\Delta$ fixed at $J = 1.67$ meV and $\Delta = 0.046$[18], which were determined from the analysis of the saturation field $H_s = 32.5$ T with the g-factor of 3.85 and the zero-field ESR gap of 0.68 meV that corresponds to the excitation gap at **Q** = (1/3, 1/3, ±1)[17, 18]. Low-energy single-magnon excitations for **Q** = (1/3, 1/3, L) are well described by LSWT with these exchange parameters. Because both the interlayer exchange interaction and the anisotropy of the exchange interaction are less than 5% of J and magnetic excitations above 1 meV are almost dispersionless along the $c^*$ direction as shown in Fig. 2b, c, we can deduce that all the excitations except the low-energy excitations near the K point can be attributed to the 2D spin-1/2 TLHAF.

**Three-stage energy structure**. The most noteworthy feature of the excitation spectrum is its three-stage energy structure. The lowest stage ($\hbar\omega < 1.6$ meV) is composed of two distinct branches of single-magnon excitations, which rise up from the K point. The middle ($1.1 < \hbar\omega < 2.4$ meV) and highest ($\hbar\omega > 2.4$ meV) stages are dispersive continua. In the spin-3/2 TLHAF CuCrO2, an excitation spectrum with such a three-stage energy structure is not observed[31]. Because the quantum fluctuation in spin-1/2 case is considerably stronger than that in spin-3/2 case, we infer that the three-stage energy structure arises from the quantum many-body effect characteristic of a spin-1/2 TLHAF.

As shown in Fig. 1a–d, a significant feature of the magnetic excitations is the two strong dispersive continua that form the middle and third stages of the excitation spectrum. The highest third stage is accompanied by a columnar continuum extending

to at least 6 meV. Figures 3a–f show constant-energy slices of the scattering intensity in the continuum range plotted in 2D reciprocal lattice space. The evolution of the scattering intensity with increasing energy is clearly observed in these figures. At intermediate energies of $\hbar\omega \sim 2.0$ meV, strong scattering occurs around the K point. With increasing energy, the position of the strong scattering shifts to the M point and the intensity at the K point decreases. We can see that the excitation continuum extends to over 8 meV.

**High-energy excitation continuum**. Figure 4a, b show the energy dependence of the excitation spectrum at the M point for **Q** = (1/2, 1/2) measured with $E_i = 7.74$ and 15.16 meV, respectively, where the scattering intensity was integrated over L. *Horizontal bars* in Fig. 4a are the energy resolution. The *inset* of Fig. 4b shows the energy dependence of the excitation spectrum for **Q** = (1/2, 1/2) measured with $E_i = 3.14$ meV. *Horizontal blue lines* in Fig. 4a, b are background level, which was estimated from the scattering intensity between 0.12 and 0.5 meV measured with $E_i = 3.14$ meV. *Small peaks* between 5 and 6 meV in Fig. 4a are extrinsic peaks originating from γ-rays emitted in the beam line. A three-stage energy structure composed of two single-magnon excitations and two excitation continua is clearly observed. For $E_i = 7.74$ meV, the widths of two single-magnon peaks are approximately the same as the energy resolution. The energy ranges of two excitation continua are much larger than the energy resolution. The third stage has a long energy tail of excitation continuum. The energy tail continues to over 10 meV, which is six times larger than the exchange interaction J.

**Discussion**

The *solid lines* in Fig. 1a, c are dispersion curves calculated by LSWT with $J = 1.67$ meV and $\Delta = 0.046$[18] on the basis of the 2D

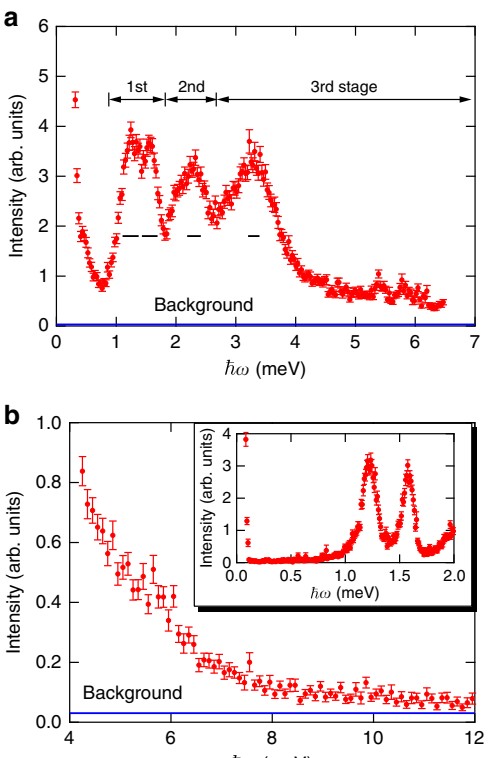

**Fig. 4** Scattering intensity at M point for Q = (1/2, 1/2) as a function of energy. **a** Three-stage energy structure composed of single-magnon excitations and two excitation continua measured at $T = 1.0$ K with $E_i = 7.74$ meV, where the scattering intensities were integrated over $L$. *Horizontal black bars* denote the energy resolution. **b** Energy tail of excitation continuum measured with $E_i = 15.16$ meV. The *inset* shows the excitation spectrum measured with $E_i = 3.14$ meV. *Horizontal blue lines* in **a** and **b** are background level, which was estimated from the scattering intensity between 0.12 and 0.5 meV measured with $E_i = 3.14$ meV as shown in the *inset* of **b**. The error of scattering intensity was calculated by the square root of absolute neutron count combined with the law of propagation of errors. The intensity and its error in this figure are given by multiplying absolute values of neutron counts and errors by the same scale factor

model described by the first term of Eq. 1. For the low-energy single-magnon excitations near the $K$ point, the spectrum becomes visually broad owing to the finite dispersion along the $c^\star$ direction. In the vicinity of the $K$ point, the lower bound of the spectrum, which corresponds to that for odd $L$ and closely approximates the spectrum at the 2D limit, coincides with the LSWT result. However, the further the wave vector moves away from the $K$ point, the more rapidly the excitation energy deviates downward from the LSWT dispersion. At the M point, the energies of lower and higher single-magnon excitations are renormalized downward by a factor of 0.69 and 0.61, respectively. This result is qualitatively consistent with the theory[21–26]. Both single-magnon branches show distinct roton-like minima at the M point, although the theory predicts that only the lowest branch shows a minimum[22, 26, 27]. The roton-like minimum is theoretically interpreted in terms of pairs of spinons characteristic of the RVB state[22, 26] or vortex excitations with fermionic character[27]. The experimental dispersion curve for the lowest branch in Ba₃CoSb₂O₉ is qualitatively in agreement with the result of series expansion approach[22, 26]. In the present experiment, we confirmed that the dispersion of single-magnon excitations is largely renormalized downward at high energies by quantum

fluctuations, while for low energies, the renormalization is small. Note that this quantum renormalization is in contrast with that observed in the spin-1/2 kagome antiferromagnet $Cs_2Cu_3SnF_{12}$ with magnetic ordering, where a uniform quantum renormalization with a **Q**-independent renormalization factor takes place[32].

Remarkable features of magnetic excitations in $Ba_3CoSb_2O_9$ are the two strong dispersive excitation continua, in which the higher energy excitation continuum extends to over the energy of 6 $J$. Because the highest energy of a single-magnon excitation is approximately equal to $J = 1.67$ meV as shown in Fig. 1a, c, the observed excitation continuum cannot be explained in terms of conventional two-magnon excitations. Recently, magnetic excitations in a spin-1/2 TLHAF were discussed from the standpoint of spin-1/2 fractionalized excitations, spinons, using a mean field Schwinger boson approach[26]. The excitation continuum in a spin-1/2 antiferromagnetic Heisenberg chain that arises from independently propagating spinons is well established[33–36]. However, it is difficult to describe the high-energy excitation continuum observed in $Ba_3CoSb_2O_9$ in terms of a two-spinon continuum in a spin-1/2 TLHAF, because the highest upper bound of continuum is approximately 2 $J$ at most[26]. At present, no theory describes the structure of the excitation continua observed in this experiment, and thus, a new theoretical framework is required. Our results show that the magnetic excitations in a spin-1/2 TLHAF include rich quantum many-body effects yet to be fully explained.

## Methods

**Sample preparation.** $Ba_3CoSb_2O_9$ powder was prepared via the chemical reaction $3BaCO_3 + CoO + Sb_2O_5 \rightarrow Ba_3CoSb_2O_9 + 3CO_2$. Reagent-grade materials were mixed in stoichiometric quantities and calcined at 1100°C for 20 h in air. $Ba_3CoSb_2O_9$ was sintered at 1200 and 1600°C for more than 20 h after being pressed into a pellet. Single crystals were grown from the melts, using a Pt crucible. The temperature at the center of the furnace was decreased from 1700 to 1600°C over 3 days. A single crystal of $10 \times 8 \times 4$ mm³ size was used in the neutron inelastic scattering experiments. The mosaicity of crystal was found to be 0.6°.

**Measurements of magnetic excitations.** Magnetic excitations in a wide momentum-energy range were measured using the cold-neutron disk chopper spectrometer AMATERAS[37] installed in the Materials and Life Science Experimental Facility (MLF) at J-PARC, Japan. The sample was mounted in a cryostat with its (1, 1, 0) and (0, 0, 1) directions in the horizontal plane. The sample was cooled to 1.0 K using a ³He refrigerator. Scattering data were collected by rotating the sample around the (−1, 1, 0) direction with a set of incident neutron energies, $E_i = 3.14$, 4.68, 7.74 and 15.16 meV. All the data were analyzed using the software suite Utsusemi[38]. At 10 K, excitation spectra shown in Fig. 1 are considerably smeared and the intensities decrease. From this result, their origin was verified to be magnetic.

**Data Availability.** All relevant data are available from the corresponding author on request.

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

## Acknowledgements

We express our sincere thanks to T. Masuda and M. Soda for their support in determining crystal orientations by the X-ray diffraction. This work was supported by Grants-in-Aid for Scientific Research (A) (No. 26247058 and 17H01142) and (C) (No. 16K05414) from Japan Society for the Promotion of Science.

## Author contributions

S.I. and H.T. grew the single crystal. S.I., N.K., H.T., S.O.-K., K.N., Sh.I. and Ke.K. performed the inelastic neutron scattering experiments. Ka.K. supervised the project. H.T. wrote the manuscript. All the authors discussed the results and contents of the manuscript.
