## [Peer Review File · Nature Communications]

Reviewers' comments:

Reviewer #1 (Remarks to the Author):

The authors provide an account of the "whole picture" of magnetic inelastic scattering in the interesting and enigmatic triangular $s=1/2$ system, $\text{Ba}_3\text{CoSb}_2\text{O}_9$. This interesting title is fully justified by the really beautiful high quality data that the paper is based on, taken on the AMATERAS chopper spectrometer at MLF, J-Parc. This data is really of the highest quality, and the authors should be commended on this - and for the general clarity of the paper they present.

The main result of this work is that there are 2 ranges of continua which are observed, and which (in the case of the higher energy range) extend in energy way beyond the normal two-spinon continuum level expected in lower dimensional spin $1/2$ magnets. This continuum scattering (referred to as broadening) was already spotted in the previous work of Ma and co-workers (authors' ref. [29]) and was considered by them to be wholly anomalous, since the kinematic conditions for two-spinon creation are violated by the strong (as they saw it) easy plane anisotropy. In fact, as far as I can tell from the literature, the easy plane anisotropy extracted from ESR is much smaller than that extracted from INS. It is interesting to note that the current paper bases its fits on the ESR value - this is not explained in the text, and this is a shame since I guess this data could have supported an independent estimate of the exchange-anisotropy.

The main problem with the paper is I guess that the authors have no theoretical model for the continua - in much the same way as Ma had no theoretical model for the continua. Ma speculated that the Hamiltonian was simplistic and that a "new theoretical framework" was needed to take this further - and there is no doubt he's right. The data presented in this paper is therefore extremely valuable in this regard - and should certainly be published. However, since the main conclusions of this work are really not original - but rather emphasise the previous conclusions of Ma and co-workers, I am sorry to say that this paper probably doesn't reach the high standards set by Nature publications.

This data should really be published as something like a PRB rapid comm. - it is interesting and instructive and the the paper is very well written, so it should have no trouble.

I would suggest that the authors avoid claims such as "Thus, the three-stage energy structure is attributed to the quantum many-body effect characteristic of a spin- $1/2$ TLHAF". This is really not substantiated. With significant out-of-plane interactions, $\text{Ba}_3\text{CoSb}_2\text{O}_9$ cannot be regarded as an archetypal model system.

Reviewer #2 (Remarks to the Author):

The authors have realized an extensive study of the magnetic excitations of $\text{Ba}_3\text{CoSb}_2\text{O}_9$ a very good experimental realization of a spin- $1/2$ triangular Heisenberg anti-ferromagnet. The results are impressive by their quality and the range of energy studied. They show that the current theoretical understanding of the excitations of this system is very insufficient. The enigma posed by the continua of high energy is today whole. As a result, these results will certainly be the source of much theoretical work.

I strongly recommend the publication in Nature Communications.

Claire Lhuillier

Reviewer #3 (Remarks to the Author):

This manuscript describes inelastic neutron spectroscopy measurements carried out $\text{Ba}_3\text{CoSb}_2\text{O}_9$,

a compound in which spin-1/2 magnetic moments are antiferromagnetically coupled on a structurally ideal triangular lattice. Neutron spectroscopy measurements on single crystal samples of this material have already been reported in References 28 and 29. One new contribution to the literature provided by this work is the extension of the inelastic spectrum measurements to considerably higher energy transfers. Beyond that, this is a fairly rare case where much of the data are measurements that have already been reported in the literature but these new results feature an improvement in statistics significant enough to provide new physics even at the qualitative level. The neutron spectroscopy results shown in Figure 1 are quite simply beautiful, and these results could significantly further understanding of a material that has attracted a lot of attention and appears to be a good model for the fundamental system of the spin-1/2 triangular lattice antiferromagnet. I think that the manuscript might be appropriate for Nature Communications with revision, though I have a few comments and questions about the analysis.

First, one reason why the statistics on these data are so impressive is because the authors have integrated the scattering over the L direction (perpendicular to the triangular lattice planes). There is plenty of evidence that the one-magnon spectrum is L-independent at $(1/2, 1/2, L)$ as shown in Figure 1(f). However the spectrum obviously disperses along L for $(1/3, 1/3, L)$ scans as shown in Figure 1(e). I'd like to see more evidence that such an aggressive L-integration is still appropriate for the higher energies as well. I suppose that Figure 1(f) shows slightly elevated scattering for energies above 1.8 meV. Is that evidence that the 'second stage' is L-independent at $(1/2, 1/2, L)$ or is it an increased background as the energy transfers approach the top of the dynamically allowed range (common on time-of-flight instruments)? Is there any evidence (likely from the $E_i=7.74$ meV data) to support the idea that the 'third stage' is L-independent for $(1/2, 1/2, L)$ or anywhere else so that one can be confident that the L-integration is valid?

The authors find a very clear splitting in the one-magnon modes, with a splitting of about 0.5 meV at the M-points. This splitting had already been hinted at by Reference 29 and is very clear in this data. This splitting is not observed in any of the modelling and I have not seen it predicted in theory. I feel that it may need a bit more explication: in particular, could this arise from an asymmetry in the system? One supposed advantage of Ba₃CoSb₂O₉ over other spin-1/2 triangular lattice antiferromagnets (such as Cs₂CuCl₄) is that it is supposed to have a perfect triangular symmetry.

In Figure 3(b) the authors emphasize finding scattering above background out to around 10 meV. This is a very slight excess though, though I think that they need to describe how the background was determined. Though I think they are probably correct that there is a magnetic contribution here: Figure 2(f) shows Q-space structure to the scattering at these high energy (though that plot could use a different color scale to emphasize this).

The authors note that they find columnar magnetic scattering stretching to high energies while continuum from fractionalized spin-1/2 spinons should stretch no higher than 2J. This is interesting and important to note. But I think they go too far with the line "These results imply that elementary excitations in a spin-1/2 TLHAF carry fractionalized spin that is smaller than 1/2." That is a very bold claim that would radically challenge a lot of current theory. I feel that it is better to let the data stand for itself rather than push this interpretation.

Reply to the criticism and comments from referee #1.

Originality of our work:

Referee #1 criticizes that our work is lack of originality, because the single-magnon and excitation continua were already reported in Ref [29]. We would not say that it is a completely fair criticism from following reasons. In Ref [29], the spectra of single-magnon excitations are indistinct, and the rotonlike minimum of higher branch was not observed at the M point. As for the excitation continuum, little is reported in Ref [29]. Two-energy stage structure of excitation continuum, its intensity map and energy range were not reported, because their energy range is lower than 3 meV. In our manuscript, we show distinct excitation spectrum including clear evidence of rotonlike minima at the M point for both magnon modes and three-stage energy structure. Because the excitation continuum reflects the characteristics of magnetic excitations, the measurements of its intensity map and energy range are essential.

Revision for this point:

- 1) 3rd paragraph of Introduction (p. 3), lines 11: we add “**that reflects the characteristics of magnetic excitations.**” After “excitation continuum”.
- 2) 3rd paragraph of Introduction (p. 3), lines 15: we add “**Little is known about the excitation continuum in Ba₃CoSb₂O₉.**”

On the magnitude of the easy-plane anisotropy:

Referee #1 asks us to explain the magnitude of the easy-plane anisotropy. As shown in our work, the dispersion relations of single-magnon excitations near the magnetic Bragg point (K point) agrees with the result of linear spin wave theory (LSWT). In Ref. [18], we observed the collective ESR mode with zero-field gap of 0.68 meV, which is consistent with the excitation gap at $\mathbf{Q}=(1/3, 1/3, \pm 1)$. The zero-field gap of the ESR mode is mainly determined by predominant intralayer exchange interaction J and anisotropy factor Δ . In Ref. [18], we evaluated Δ to be 0.046, using an analysis that is equivalent to LSWT. Also we can see in Fig. 1f that the excitation gap at $\mathbf{Q}=(1/3, 1/3, \pm 1)$ is well described with $\Delta = 0.046$.

Revision for this point:

- 1) 1st paragraph of the first subsection of Results (p. 5), line 14: we add “**, which were determined from the analysis of the saturation field $H_s = 32.5$ T with the g -factor of 3.85 and the zero-field ESR gap of 0.68 meV that corresponds to the excitation gap at $\mathbf{Q} = (1/3, 1/3, \pm 1)$ ^{17,18}. Low-energy single-magnon excitations for $\mathbf{Q} = (1/3, 1/3, L)$ are well described by**

LSWT with these exchange parameters.” After “ $J = 1.67$ meV and $\square\square = 0.046$ ”¹⁸;

On the last criticism and interlayer exchange interaction:

Referee #1 criticized that our statement “the three-stage energy structure is attributed to the quantum many-body effect characteristic of a spin- 1/2 TLHAF” is really not substantiated. Also referee #1 stated With significant out-of-plane interactions, Ba₃CoSb₂O₉ cannot be regarded as an archetypal model system. The second statement would be misunderstanding. Because the interlayer exchange interaction is less than 5% of J and the experimentally observed magnetization process including a 1/3 plateau is in excellent agreement with theory based on the Heisenberg model, it is certain that Ba₃CoSb₂O₉ closely approximates the ideal spin-1/2 TLHAF. It is clear from Figs. 1g and h that the excitations above 1 meV is independent of L , which demonstrates that the exchange network is two-dimensional (2D).

Revision for this point:

- 1) 1st paragraph of the second subsection of Results (p. 5), line 6: we changed a sentence “**Thus, the three-stage energy structure is attributed to the quantum many-body effect characteristic of a spin-1/2 TLHAF.**” to “**Because the quantum fluctuation in spin-1/2 case is considerably stronger than that in spin-3/2 case, we infer that the three-stage energy structure arises from the quantum many-body effect characteristic of a spin-1/2 TLHAF.**”
- 2) 1st paragraph of the first subsection of Results (p. 5), line 4 from the end line: we add a sentence “**and magnetic excitations above 1 meV are almost dispersionless along the c^* direction as shown in Figs. 1g and h,**” between “... less than 5% of J ” and “we can deduce that ...”

Reply to the criticism and comments from referee #3.

Excitation spectrum along the c^* direction:

Referee #3 asks us to show that not only the single-magnon excitations but also excitation continuum are independent of L above 1 meV. We show in Fig. 1h the scattering intensity along $(1/2, 1/2, L)$ measured with $E_i=7.74$ meV, which indicates that all the magnetic excitations are dispersionless along c^* direction above 1 meV.

Revision on this point:

- 1) We show the scattering intensity along $(1/2, 1/2, L)$ measured with $E_i=7.74$ meV in Fig. 1h.

On the splitting of magnon modes at the M point:

Referee #3 asks us to explain the reason why the single-magnon mode splits into two modes at the M point. This is not the splitting but the downward renormalization of excitation energies of two different modes. The excitation energies of 1.8 and 2.6 meV calculated by LSWT for the lower and higher excitation modes (white lines in Fig. 1a and c) decrease to 1.2 and 1.6 meV, respectively, owing to quantum many-body effect, as theory predicts Refs. [21-26].

Revision on this point:

- 1) 1st paragraph of Discussion (p. 7), line 8: We add a sentence “**At the M point, the energies of lower and higher single-magnon excitations are renormalized downward by a factor of 0.69 and 0.61, respectively.**”

On the background level:

Referee #3 asks us to describe how the background was determined. The background level was estimated from the scattering intensity between 0.12 and 0.5 meV measured with $E_i=3.14$ meV for $\mathbf{Q}=(1/2, 1/2)$.

Revision on this point:

- 1) We show the energy dependence of excitation spectrum for $\mathbf{Q}=(1/2, 1/2)$ measured with $E_i=3.14$ meV in the inset of Fig. 3b.
- 2) In the last paragraph of Results, we add a sentence “**The inset of Fig. 3b shows the energy dependence of the excitation spectrum for $\mathbf{Q} = (1/2, 1/2)$ measured with $E_i = 3.14$ meV. Horizontal blue lines in Figs. 3a and b are background level, which was estimated from the scattering intensity between 0.12 and 0.5 meV measured with $E_i=3.14$ meV.**”

On the last comment:

Referee #3 points out that our statement “**These results imply that elementary excitations in a spin-1/2 TLHAF carry fractionalized spin that is smaller than 1/2.**” is overstatement.

Revision on this point:

- 1) The last sentence of abstract (p. 2): we change the sentence “**Our results strongly suggest the necessity of using small fractionalized spin excitations to describe the high-energy excitations in spin-1/2 TLHAFs.**” to “**Our results strongly indicate the necessity of a new theoretical framework.**”
- 2) 2nd paragraph of Discussion (p. 8): we delete a sentence “**The present experimental result**

strongly suggests that the excitation continuum in a spin-1/2 TLHAF is composed of multiple excitations of fractionalized spin excitations that are smaller than 1/2.” We add to the last sentence “, and thus, a new theoretical framework is required. Our results show that the magnetic excitations in a spin-1/2 TLHAF include rich quantum many-body effects yet to be revealed.”

- 3) We delete the last paragraph of Discussion.

REVIEWERS' COMMENTS:

Reviewer #1 (Remarks to the Author):

The authors have improved their submission by removing their over-generalised claims of fractionalised excitations, and the generality of their 3-stage excitation structure as being indicative of a spin 1/2 TLHAF. I guess at the very least, we would need one more example to claim such generality.

Still I worry that the authors' claim of generality in their reply betrays a tendency to overplay their hand. They state that:

"Because the interlayer exchange interaction is less than 5% of J and the experimentally observed magnetization process including a 1/3 plateau is in excellent agreement with theory based on the Heisenberg model, it is certain that Ba₃CoSb₂O₉ closely approximates the ideal spin-1/2 TLHAF." The word "certain" has no place in science. And the phrase "closely approximates" is subjective. However, the authors have now shown evidence of the lack of dispersion along L in the new revision, and that does give the reader some appreciation as to the dimensionality of the interactions.

The paper is free from errors, presents exceptional quality INS data, and deserves publication. I leave it to the editors to decide whether the paper presents significantly new results. I do not say (nor did I before) that the paper was unoriginal. But rather that the main conclusions of the article are similar (and now, exactly the same) as those of the paper of Ma (their ref [29]). I of course accept that the data they present is of much higher quality and wider energy range, but the conclusions appear to be the same.

Reviewer #3 (Remarks to the Author):

The authors have appropriately responded to all specific questions and criticisms that were raised during the first round of review. I think that the manuscript has been strengthened significantly by the peer review process. As currently written the paper is technically sound and appropriate for publication.

The remaining question is whether or not the paper is of sufficient impact to be well-suited for Nature Communications. My opinion during the first review was that, while the presented experiments are mostly not new, the improvement in statistics presented here is of sufficient magnitude to qualitatively further understanding of the material. In this way the results are novel even if the experiments are not.

Reply to the criticism and comments from referee #1.

Reviewer #1 criticizes that a word “certain” in our reply “Because the interlayer exchange interaction is less than 5% of J and the experimentally observed magnetization process including a 1/3 plateau is in excellent agreement with theory based on the Heisenberg model, it is certain that $\text{Ba}_3\text{CoSb}_2\text{O}_9$ closely approximates the ideal spin-1/2 TLHAF.” has no place in science. Reviewer #1 also criticizes that "closely approximates" is subjective.

Revision for this point:

- 1) We correct our statement in this reply as follows: “Because the interlayer exchange interaction is less than 5% of J and the experimentally observed magnetization process including a 1/3 plateau is in excellent agreement with theory based on the Heisenberg model, we can deduce that $\text{Ba}_3\text{CoSb}_2\text{O}_9$ can be described as a spin-1/2 TLHAF.”
- 2) 2nd paragraph of Introduction (p. 2), line 6-8: we revised the sentence “The quantum magnetization process has been quantitatively verified by high-field magnetization measurements on $\text{Ba}_3\text{CoSb}_2\text{O}_9$, which closely approximates the ideal spin-1/2 TLHAF^{17,18}.” to “The quantum magnetization process has been quantitatively verified by high-field magnetization measurements on $\text{Ba}_3\text{CoSb}_2\text{O}_9$, which is described as a spin-1/2 TLHAF^{17,18}.”